# In Vitro High-Throughput Toxicological Assessment of Nanoplastics

**DOI:** 10.3390/nano12121947

**Published:** 2022-06-07

**Authors:** Valentina Tolardo, Davide Magrì, Francesco Fumagalli, Domenico Cassano, Athanassia Athanassiou, Despina Fragouli, Sabrina Gioria

**Affiliations:** 1Smart Materials, Istituto Italiano di Tecnologia, Via Morego, 30, 16163 Genova, Italy; valentina.tolardo@iit.it (V.T.); athanassia.athanassiou@iit.it (A.A.); despina.fragouli@iit.it (D.F.); 2Department of Informatics, Bioengineering, Robotics and Systems Engineering, University of Genova, Via All’ Opera Pia, 13, 16145 Genova, Italy; 3European Commission, Joint Research Centre (JRC), 21027 Ispra, Italy; davide.magri.mail@gmail.com (D.M.); francesco-sirio.fumagalli@ec.europa.eu (F.F.); domenico.cassano@sns.it (D.C.)

**Keywords:** nanoplastics, polyethylene terephthalate, polycarbonate, laser ablation, in vitro assays, high content screening, cytotoxicity, nanotoxicology

## Abstract

Sub-micrometer particles derived from the fragmentation of plastics in the environment can enter the food chain and reach humans, posing significant health risks. To date, there is a lack of adequate toxicological assessment of the effects of nanoplastics (NPs) in mammalian systems, particularly in humans. In this work, we evaluated the potential toxic effects of three different NPs in vitro: two NPs obtained by laser ablation (polycarbonate (PC) and polyethylene terephthalate (PET1)) and one (PET2) produced by nanoprecipitation. The physicochemical characterization of the NPs showed a smaller size, a larger size distribution, and a higher degree of surface oxidation for the particles produced by laser ablation. Toxicological evaluation performed on human cell line models (HePG2 and Caco-2) showed a higher toxic effect for the particles synthesized by laser ablation, with PC more toxic than PET. Interestingly, on differentiated Caco-2 cells, a conventional intestinal barrier model, none of the NPs produced toxic effects. This work wants to contribute to increase knowledge on the potential risks posed by NPs.

## 1. Introduction

Although the production of the first synthetic plastic, Bakelite, occurred in 1907, it was not until the late 1950s that the plastic industry initiated its rapid growth. Over the next 70 years, annual global production increased, reaching 367 million metric tons in 2020 [1]. While plastic is a unique material in terms of its versatility, cost-effectiveness and resistance, these same properties combined with the intensive consumption and rapid disposal of plastic products have caused its accumulation in the environment [2,3]. Nowadays, it has been estimated that 0.1% of the total global production (>8.3 billion metric tons) has reached the ocean as plastic marine debris [4,5,6] and, at current rates, by 2040, it will increase of 2.6-fold [7]. The long persistence due to the slow degradation of plastic wastes [8] causes significant environmental and health concerns. Plastic floating in seawater is exposed to photo-oxidation, which, in combination with microbial action and mechanical wear, fragments into small debris that, as shown by various studies, can reach sizes of less than 1 µm and are termed nanoplastics (NPs) [9,10].

Plastic litter of micro and nano-size may impact the base of the ocean food chain, causing potential damage to the entire trophic chain, including humans [11,12,13,14]. Although the effects of plastics in aquatic systems are widely documented [15,16], the potential impact of NPs on humans is still unclear [17] as, to date, there are still limited studies assessing the potential toxic effects and biological interactions of NPs in mammalian systems and mainly performed using polystyrene nanobeads [18,19,20,21,22,23]. 

Only recently, ongoing efforts have been made to improve the representativeness of NPs samples and to investigate the fate and behavior in the environment, as well as the toxicity on living organisms [24,25,26,27]. Magrì et al. [28,29] have described a top–down approach to obtain PET NPs, based on the laser ablation of PET films in water. In comparison to the NPs chemically synthetized, the morphological and surface chemistry properties of the “as formed” PET NPs, obtained by laser ablation, are more similar to the NPs expected to be found in the marine environment [30]. A key point of this top–down approach is that the in-water, laser-formed NPs take into account the complexity of the degradation of plastics in the marine environment, which may lead to the release of chemical species and side products from the polymer itself [31,32,33]. In fact, it has been demonstrated that the photo-degradation of the polymers results in hazardous products associated with the reaction between polymers, water and UV light [31,34,35]. 

In this work, we assessed in vitro the potential toxic effects of NPs largely used in the food packaging industry or for beverage bottles production. As ingestion represents the main entry route for NPs to end up in the human body [12,36,37], Caco-2 and HepG2 cells were selected as in vitro models representative of the intestinal epithelial barrier and of the liver, respectively. 

As there is still a large knowledge gap in understanding the effects of the most common NPs on human health, this study wants to contribute to generating toxicity data to increase the knowledge on the potential risks posed by NPs.

## 2. Materials and Methods

### 2.1. Physical and Morphological Characterization of NPs

#### 2.1.1. NPs Fabrication 

For the synthesis of PC and PET1 NPs, pulsed laser ablation on commercial PC and PET films (Goodfellow Cambridge Ltd., Ermine Business Park, Huntingdon, England PE29 6WR) was performed using a nanosecond (ns) KrF excimer laser (Coherent-CompexPro 110, Coherent Europe B.V. Kanaalweg, Utrecht, The Netherlands). The films, immersed in 8 mL of Milli-Q^®^ water, were exposed to laser irradiation (KrF excimer laser, wavelength: 248 nm, pulse duration: 20 ns, repetition rate: 10 Hz, number of pulses: 50, energy *fluence*: 2.8 J/cm^2^ for PC NPs and 4.5 J/cm^2^ for PET NPs). The laser was coupled with a micromachining apparatus (Optec-MicroMaster, Optec S.A., ZAE Le Crachet, Avenue des Nouvelles Technologies, 53 B-7080, Frameries, Belgium) in order to irradiate an array of 4 cm^2^ in automated mode. Details of the synthesis procedures are described in previous work [28]. For PET2 NPs, PET powder from milled plastic water bottles was treated with hydrochloric acid 50% *v/v* in Milli-Q^®^ water. Dried powder was dissolved in 1,1,1,3,3,3-hexafluoro-2-propanol (HFIP) at the concentration of 2 mg/mL; then, PET2 NPs were developed through a modified flash nanoprecipitation protocol [38]. Briefly, 5 mL of HFIP solution of PET (2 mg/mL) was added dropwise to 50 mL of a water solution of sodium cholate 10 mg/mL in mild stirring. HFIP was removed through rotary evaporation, and then, NPs were washed twice with Milli-Q^®^ water to remove the excess of sodium cholate. For the washing steps, centrifugation was performed at 13,000× *g* for 3 min. The NP suspension in Milli-Q^®^ water was stored at 4 °C. In addition, an aliquot of PC and PET1 NPs were further filtrated using Millipore Amicon™ Ultra Centrifugal Filter Units (cut-off 30 kD, Cat number UFC5030) at 10,000× *g* for 30 min at 20 °C in order to separate the NPs dispersant by the particulate fraction. In the case of PET2, a solution of sodium cholate hydrate 1 mg/mL was used as respective “dispersant”.

#### 2.1.2. Hydrodynamic Diameter (DH) Distribution and Morphological Characterization 

The analyses of the hydrodynamic diameter (D_H_) by Dynamic Light Scattering (DLS) were performed using a Zetasizer Nano ZS spectrometer (Malvern Instruments, Malvern WR14 1XZ, UK). For the measurements, all NPs were dispersed in Milli-Q^®^ water to a final concentration of 30 µg/mL. The size distribution results were expressed as an average of three consecutive measurements carried out at 25 °C. NPs morphology was evaluated by a transmission electron microscope (TEM) (JEM-1011, JEOL, Tokyo, Japan) with a thermionic source (W filament) and accelerating voltage of 100 kV. For their characterization, the polymeric particles dispersions were drop casted on ultrathin carbon layered Cu grids (CF300-CU-UL) (Electron Microscopy Science, Hatfield, Pennsylvania) at room temperature. 

#### 2.1.3. X-ray Photoelectron Spectroscopy (XPS) Analysis

XPS measurements were performed with an Axis Ultra spectrometer (Kratos Analytical, Manchester, UK), using a Kα Al monochromatic source (hν  =  1486.6 eV) operating at 150 W (15 kV, 10 mA) and an X-ray spot size of 400 × 700 μm^2^ in hybrid electromagnetic lens configuration mode. The residual pressure of the analysis chamber during the analysis was less than 8 × 10^−9^ Torr. For each sample, both survey spectra (0–1150 eV, pass energy 80 eV) and high-resolution spectra (pass energy at 40 eV) were recorded. The surface charge was compensated by a magnetic charge compensation system, and the energy scale was calibrated by setting the C 1 s hydrocarbon peak to 285.00 eV in binding energy. The data were acquired using Vision2 software (Kratos Analytical, Stretford, UK), and the analysis of the XPS peaks was carried out using a commercial software package (CasaXPS v2.3.18PR1, Casa Software, Ltd., Teignmouth, UK). Peak fitting was performed with no preliminary smoothing. Symmetric Gaussian–Lorentzian (70% Gaussian and 30% Lorentzian) product functions were used to approximate the line shapes of the fitting components after a 3-parameter Tougaard-type background subtraction. NP samples dispersed in Milli-Q^®^ water were drop-casted on both clean Teflon substrates and Si wafers. The use of Teflon substrate allows minimizing the uncertainties due to adventitious hydrocarbons contamination. 

### 2.2. Cell Culture Conditions 

Two in vitro cell lines have been used in this study. Human hepatocellular carcinoma cells (HePG2), recommended by several standards to assess nanoparticles toxicity [39,40], were purchased from LGC standards (Cat HB-8065™, lot number 62591368) and cultured in complete culture medium composed of Roswell Park Memorial Institute (RPMI) 1640 medium supplemented with 10% (*v*/*v*) Fetal Bovine Serum (FBS, North America Origin), 0.5% (*v*/*v*) penicillin/streptomycin and 6 mM L-glutamine. Human colon adenocarcinoma Caco-2 cells, a widely used cell culture model, were purchased from Sigma-Aldrich (Cat 86010202, lot number 09C004) and cultured in complete culture medium, composed of Dulbecco’s Modified Eagle Medium (DMEM) high glucose (4500 g/L) supplemented with 10% (*v*/*v*) Fetal Bovine Serum (FBS, North America Origin), 0.5% (*v*/*v*) penicillin/streptomycin, 12 mM L- glutamine and 1% (*v*/*v*) non-essential amino acids. All cell culture reagents were purchased from Life Technologies, Italy. For routine culture, cells were maintained in a sub-confluent state under standard cell culture conditions in a humidified incubator (37 °C, 5% CO_2_, 95% humidity) (Heraeus Thermo Fisher^®^, Houthalen, Belgium). For HCS experiments, RPMI and DMEM were used without phenol red.

### 2.3. Cytotoxicity Evaluation by LDH and MTT Assay

NPs cytotoxicity was firstly evaluated in vitro using two colorimetric assays, the lactate dehydrogenase (LDH) release and the 3-(4,5-dimethylthiazol-2-yl)-2,5-diphenyltetrazolium bromide (MTT) reduction test, on HepG2 cells [39,40]. Cells were plated in 96-well cell culture plates (Corning Inc., Corning, NY, USA) at a density of 5 × 10^4^ cells/well and allowed to adhere for 24 h. Cells were then exposed to the NPs (1, 10, 20, 40 or 80 µg/mL) for 24 or 48 h. A negative control (medium) and a positive control (Triton 0.1%) were included. For each type of NP dispersion (80 μg/mL), the dispersant was also assessed and tested. At the end of the exposure time, 50 µL of supernatant was transferred in a new plate for measuring LDH release, using the BioVision LDH-cytotoxicity colorimetric kit (cat. K311, BioVision, Inc., Milpitas, CA, USA) according to the manufacturer’s instruction. Cell viability was evaluated using MTT 3-(4,5-dimethylthiazol-2-yl)-2,5-diphenyltetrazolium bromide (Sigma-Aldrich, Inc., Milan, Italy), which was added to the cells in fresh complete culture medium at a final concentration of 250 µg/mL. After 4 h of incubation at 37 °C, the supernatant was removed, the precipitated formazan crystals were dissolved in 200 µL RPMI (Sigma-Aldrich, Inc., Milan, Italy) followed by 50 µL of glycine buffer (0.1 M glycine with 0.1 M NaCl in MilliQ water). The absorbance was measured at 490 nm and 570 nm for LDH and MTT assay, respectively, by the EnSpire^®^ Multimode plate reader (Perkin Elmer) using a reference wavelength of 680 nm. Data are expressed as percentage of total LDH release and as percentage of mitochondrial activity and reported as mean ± SD. Three independent experiments were performed, and each condition was run in triplicate.

### 2.4. High Content Screening (HCS) 

#### 2.4.1. Cell Culture and NPs Exposure

HepG2 and Caco-2 cells were plated in Corning^®^ 96-well black plates at a density of 1 × 10^4^ cells/well and 5 × 10^3^ cells/well, respectively, allowing 24 h for adhesion before treatments. For experiments with differentiated Caco-2 cells, a seeding density of 8 × 10^4^ cells/well was used. For differentiation, cells were maintained in phenol red free complete medium, and the medium was replaced three times/week until complete differentiation at day 21 [41]. For experiments, cells were exposed to NPs (10, 20, 40 or 80 µg/mL) for 24 or 48 h. Negative control (medium), positive control (Valinomycin 900 nM) and NPs dispersants (as previously described in par. 3) were also included in the study. Three independent biological experiments have been performed and, for each experiment, three technical replicates have been made.

#### 2.4.2. HCS Assay: Incubation of Fluorescent Staining and Imaging

The Mitochondrial Health Kit (Invitrogen, Thermo Fisher, Houthalen, Belgium) was used to stain the nuclei (Hoechst 33342), the cell membrane (Image-iT^®^ DEAD Green™ viability stain) and the mitochondrial membrane (MitoHealth stain). Briefly, cells were exposed to NPs at the concentrations of 10, 20, 40 or 80 µg/mL and compared to negative and positive controls. After staining and cell fixation according to the manufacturer’s instructions, the IN Cell analyzer 2200 (GE Healthcare, Boston, MA, USA) was used to acquire images in high-throughput mode. Nine fields per well were imaged with a 10× objective, and an average of 2000 objects/field was counted for the analysis. 

#### 2.4.3. HCS Data Analysis

An IN Cell Developer Toolbox 1.9.2 from GE Healthcare (Boston, MA, USA) was used for data analysis. Data were normalized to the negative control (untreated cells) and expressed as an average of three independent experiments (each run in technical triplicate) ± SD. Data were graphically represented using Origin Pro version (2018).

### 2.5. Electric Cell-Substrate Impedance Sensing (ECIS) Technology

Differentiated Caco-2 cells were obtained after plating Caco-2 cells in 96-wells ECIS disposable arrays at a density of 8 × 10^4^ cells/well and maintaining for 21 days until complete differentiation, as reported in Section 2.4.1. The Electric Cell-substrate Impedance Sensing (ECIS, Applied BioPhysics, Inc., NY, USA) was used to monitor in real time electrical changes. At complete cellular differentiation, the medium was replaced with fresh medium (controls) or with medium containing NPs at the concentrations of 10, 20, 40 or 80 µg/mL. Dispersants were also included as for the highest NP dose tested (80 μg/mL). Each experimental condition was tested in triplicate. After 48 h, treatments were removed, and cells were maintained for 12 days in fresh complete culture medium for recovery, during which medium was regularly replaced every 3 days. Cells were then exposed to a second treatment as previously described. As last, Triton (0.1%) was added to each well and used as reference (positive control). The data were collected and analyzed by ECIS Zθ software (ECIS^TM^ v1.2). Origin Pro version (2018) was used for data representation.

### 2.6. Statistical Analysis

Data are reported as means ± standard deviations (SD) of three independent experiments, each of them run in technical triplicate. Statistical significance of differences is indicated by *p* values, which are calculated using one-way ANOVA, Origin 2018. EC_50_ values were calculated using Graph Pad Prism by fitting a sigmoid curve (log (inhibitor) vs. normalized response—variable slope). 

In addition, a two-way ANOVA has been performed on HCS data to monitor simultaneously mitochondrial activity and dead intensity in order to correlate simultaneously the NPs effects.

## 3. Results

### 3.1. NP Physicochemical Characterization

Three NPs were investigated in this study. PC and PET1 NPs were obtained by laser ablation, whereas PET2 NPs were chemically synthetized. The NPs formed by laser ablation were previously characterized in terms of shape, size distribution, surface and colloid chemistry [28]. Batch-mode Dynamic Light Scattering (DLS) was performed to investigate the particle size distribution (PSD) and sample polydispersity. Data are summarized in Figure 1a. Results show that PC has a hydrodynamic diameter (D_H_) of 47.4 ± 3 nm and a polydispersity index (PDI) of 0.190, PET has a D_H_ of 57.7 ± 9 nm with a PDI of 0.196; whereas PET2 NPs D_H_ is of 88.9 ± 2 nm with a PDI of 0.080. Batch-mode DLS was repeated in time during the whole study to monitor the NPs’ stability. Data showed that NPs remained stable for more than six months in Milli-Q^®^ water. In addition, all the NPs were stable in complete culture medium for up to 48 h (data not shown). 

In terms of size, TEM images (Figure 1b–d) agree with DLS results, showing an almost spherical shape for all the materials, with a more irregular surface in the case of the laser ablated NPs. 

Moreover, the surface chemistry characterization analyzed by XPS showed for PC and PET1 NPs a net negative charge and a high oxidation degree with the exposure of carboxyl and hydroxyl groups on the surface, in contrary to the pristine material (Appendix A and Appendix A). The ratio of the oxidized carbon bond with the aliphatic/aromatic C-C bonds increases in the laser-ablated NPs, especially in the case of PC NPs, compared with the starting material. These relative abundances of different oxidized carbon bonds on the NPs surface qualitatively reflect oxidation chemistry, measured using a similar method, on plastic NPs proxies obtained via mechanical milling and fractionating environmental plastic samples [27]. In PC and PET1, a new oxidized functionality (C=O bond) is exposed, and the satellite shakeup features (related to delocalized C-C bonds in aromatic rings) decrease in PET1 NPs (Figure 2). For PET2 NPs, a negative charge and less relevant oxidation on the surface than the laser ablated NPs were found. In addition, it was observed that the PET2 NPs’ surface chemistry resembles more closely the properties of the stoichiometric materials. The chemical synthesis did not introduce additional oxidized species on the PET2 NPs’ surface (Appendix A and Appendix A). After separating the NPs from their liquid medium by an ultrafiltration column, the NPs dispersant was analyzed by NMR. In the case of PC NPs, preliminary data indicate the presence of low molecular weight intermediates, such as monomers or oligomers, which are possibly attributed to the photo-degradation by-products of the polymer itself.

### 3.2. Toxicological Assessment 

The NPs’ potential toxicity was initially assessed by two colorimetric assays (LDH release and MTT reduction test) on HePG2. Cells were exposed to NPs at different concentrations spanning from 10 to 80 µg/mL for 24 or 48 h. In addition, the respective liquid media (dispersants) were also tested. LDH results are shown in Appendix A. Data clearly indicate that the LDH method used is not suitable to assess the toxicity of the NPs under investigation. This is probably due to interferences between the NPs and the LDH assay components. 

MTT results on HePG2 cells exposed to the different NPs are reported in Figure 3. Data indicate a significant toxicity of the PC dispersant at both time points considered. A reduction in cell viability for the PC dispersant was also confirmed by morphological observation (data not shown). Statistical analysis did not indicate significant toxicity for PC NPs at all doses and time points tested, although a high standard deviation was observed among replicas, and rather, a higher mitochondrial activity was found in cells exposed at the lower doses (1, 10 and 20 µg/mL) for 24 h. 

PET1 and PET2 resulted toxic only at the highest dose tested and only at the 24 h time point, which is in line with morphological observation (Appendix A). No significant toxicity was also observed in the case of the respective dispersants. No morphological changes were evident in the case of PET2 exposed cells at all doses tested (Appendix A). 

### 3.3. High-Content Screening (HCS) 

To overcome the limitations of the classical colorimetric methods that we observed in the case of LDH assay, as well as the high variability observed for MTT experiments, cytotoxicity was also evaluated by HCS using two in vitro cell models. For this, the nuclei, the cell membrane and the mitochondrial membrane were stained in order to evaluate changes in cell viability, cell membrane permeability and mitochondrial membrane potential, respectively. Figure 4 shows HePG2 and Caco-2 cells’ viability after exposure to the NPs at the different doses or to the dispersant. Data indicate a significant toxicity of PC and PET1 dispersants at both time points considered; no toxicity was observed in the case of the PET2 dispersant. A morphological observation of HePG2 and Caco-2 cells confirms the toxic effect of PC and, to a lesser extent, PET1 dispersants already after 24 h exposure, whereas no effect was observed in the case of the PET2 dispersant (data not shown). The reduction in cell viability was also found to be significant for PC and PET1 NPs at nearly all doses tested, with PC resulting more toxic than PET1 at both time points tested, which is in line with morphological observation. PET2 did not affect significantly cell viability, except for a reduction observed at 48 h at the highest dose tested on HePG2 cells. No evidence of morphological changes was observed for PET2-exposed cells at all doses tested and at both time points considered (data not shown). Table 1 reports the EC_50_ values for each type of NPs at 24 and 48 h. Despite the low toxicity of PET2 NPs, the corresponding EC_50_ is also indicated. 

Results of the effects on the mitochondrial activity are reported in Figure 5. A reduction in the mitochondrial activity was evident for HePG2 cells exposed to PC NPs at all time points analyzed, whereas on Caco-2 cells, PC NPs were significantly toxic only at the highest dose tested. To a lesser extent, PET1 NPs also synthetized by laser ablation showed a significant reduction in mitochondrial activity at all doses tested on HePG2 cells. In the case of PET2 NPs, a reduction in mitochondrial activity was observed on HePG2 cells at 80 µg/mL, at both time points, and at 40 µg/mL only at 48 h. Again, the reduction in mitochondrial activity was more limited when PET2 NPs were exposed to Caco-2 cells. The NPs dispersant was found for HePG2 to be significantly toxic in the case of PC NPs at both time points considered, whereas PET1 NPs dispersant was found to be significantly toxic only at 48 h. The PET2 dispersant was not toxic under all conditions tested. 

Data on nuclear size and nuclear intensity were also extrapolated. No notable changes were observed for all NPs and doses tested for both cell lines and at all time points considered (Appendix A). 

For the statistical comparison of the data, the two-way analysis was used. Two different parameters related to cell viability have been monitored simultaneously: mitochondrial activity and cell membrane damage. Plotting the effect of the dead intensity against the one onmitochondrial activity allows us to correlate simultaneously the NPs’ effects to both parameters. Figure 6 reports the case of HePG2 cells. As expected, the negative and positive controls are placed on the two opposite sides in the graph, whereas the different NPs concentrations are distributed with a dose–response trend between these extremes, especially for PC, indicating an increase in the toxicity at increasing NPs concentrations. PET2 had the less toxic NPs. The two-way analysis for Caco-2 cells is reported in Figure 7. The data show that PC NPs at the highest dose tested (80 µg/mL) have a significant toxicity with a strong effect on membrane damage and loss of cell mitochondrial activity. PET1 and PET2 treatments have no or more limited effects on mitochondrial activity and cell membrane integrity compared to PC NPs. Rather, at low NPs doses, the mitochondrial activity was found to be higher compared to the control. 

In addition, HCS was also performed on differentiated Caco-2 cells. The results are shown in Figure 8 and Figure 9. In the case of differentiated Caco-2 cells, no toxic effect on cell viability and mitochondrial activity was found for all NPs under the conditions tested. Respective NPs dispersants were also assessed and shown to be not toxic (data not shown). Data of undifferentiated Caco-2 cells are also reported for better comparison.

### 3.4. Electric Cell-Substrate Impedance Sensing (ECIS) Technology

To evaluate NPs’ toxicity at long-term exposure, electric cell–substrate impedance sensing (ECIS) measurements were performed on differentiated Caco-2 cells. Cells were exposed twice to NPs for 48 h with a recovery period of 12 days between treatments. No changes in electrical impedance were reported during the whole time monitored for all doses tested or when exposed to the dispersants (data not shown). Results indicate that the exposure of monolayer of differentiated Caco-2 cells to the NPs or its dispersants does not cause any significant cell damage, confirming barrier integrity. Triton (0.1%) was used as positive control and worked as expected (reduction in the electrical impedance). Figure 10 reports the results obtained for the highest dose tested (80 g/mL).

## 4. Discussion

Plastics have enormously impacted every aspect of our daily life, and its use is growing year by year [42]. As plastic wastes are persistent environmental pollutants, with the majority ending up in the marine environment, plastic materials are exposed to degradation processes (thermal, physical, thermo-oxidative and photo-degradation, among others) and could break down into smaller pieces even reaching the nano size range. These small pieces, known as nanoplastics (NPs), have posed significant environmental and health concerns.

While NPs have been widely studied in the context of the marine environment, their effects on mammals, particularly humans, are still unclear due to the limited data available. Furthermore, most of these studies were performed using polystyrene beads, as they are easy to be synthesized in the nano size, based on the nanoprecipitation of commercial solutions, not representing the most common commercial used plastics.

Toxicological assessment of different polymeric NPs (polystyrene, polyethylene and polyvinyl chloride) shows no effects on cell viability and mitochondrial activity [43,44]; however, there are not many studies conducted on different types of NPs, and they are often limited to commercial materials such as polystyrene. In this work, we assessed in vitro the potential toxic effects of three different NPs: two NPs obtained by laser ablation (PC and PET1), which should be more similar to the NPs expected to be found in the marine environment, and one NPs produced by chemical synthesis (PET2). We selected PC and PET NPs, due to the worldwide spread of these polymers in different applications, especially in the food sector as packaging material or as plastic bottles containers.

All the material was characterized in terms of size, surface chemical composition and stability. As shown in previous works [28], the pulsed laser ablation of solid polymeric films in water results in the formation of NPs with similar characteristics as the ones expected to be found in the environment, in terms of surface and shape irregularity, broad size distribution and chemistry. Focusing on their nanometric fraction, the plastic fragments produced have an approximatively spherical shape, a higher polydispersity, a size distribution with a D_H_ of ca. 50 nm and a jagged surface. Colloidal PET2 NPs are more monodispersed and are slightly bigger, while their surface chemistry is indicative of the PET chemical composition, contrary to the more oxidized character of the laser-ablated NPs. The monodispersity, spherical shape, smooth and homogeneous surface are the main properties of chemically synthesized NPs [26,45], which are generally used as the model [46], although they are not representative of the NPs present in the environment. In fact, the natural degradation of plastic results in the formation of NPs with physical irregularity and with non-uniform surface chemistry depending on various environmental factors [32,47,48]. Moreover, the thermo-oxidative and photo-oxidative pathways of PC and PET in water have been demonstrated to induce the formation of products with characteristic exposure of oxidized functional end-groups on their surface [32,47]. These groups are, indeed, present in the chemical composition of the surface of the PC and PET1 NPs synthetized by laser ablation, while they are not observable for PET2. The photo-degradation of the PC NPs has been demonstrated to be also rich of other sub-products of the degradation itself [47]. For this reason, the dispersant of the NPs after separation was also evaluated for its toxicological potential.

The NPs toxicological assessment was performed on in vitro models, which allows the identification of the hazardous potential of xenobiotic materials. As ingestion represents the main entry route for NPs to end up in the human body [49], Caco-2 and HepG2 cells were used as in vitro models representative of the intestinal epithelial barrier and of the liver, respectively.

Toxicological assessment was initially performed according to ISO guide ISO 10993-22 (*Biological evaluation of medical devices—Part 22: Guidance on nanomaterials*) [39], which indicates the LDH assay and the MTT assay as the standard methods to address the biological evaluation of nanoparticulate materials. However, LDH assay was not applicable in the case of the NPs tested due to possible interference of the nanomaterials with the reagents or with the assay readout [50,51,52,53]. On the contrary, MTT assay showed to work despite the high variability across replicates.

Considering that several studies have been published reporting the limitations of these classical colorimetric methods for the evaluation of the toxicity of nanoparticles [51,52,53,54], we also applied HCS, using fluorescence-based methods, to assess NPs’ potential toxic effects. The advantage of HCS is the ability to analyze simultaneously multiple parameters; in particular, we focused on assessing changes in cell viability, cell membrane permeability, mitochondrial membrane potential and nuclear morphology.

By HCS, the NPs synthetized by the laser ablation approach show higher toxicity with respect to colloidal NPs, with PC being more toxic than PET1. NPs synthesized by the laser ablation approach exhibit increased polydispersity characterized by very small (<50 nm) subpopulations of fragments. The presence of these very small fragments resulting from the fragmentation of polymer molecules induced by laser ablation and not by colloidal synthesis could be linked to the toxicity [55,56,57,58]. Moreover, PC NPs show a stronger effect on cell viability, mitochondrial activity and on cell membrane damage than laser ablated PET1, despite having a similar polydispersity.

When analyzing the effect of the water dispersant, after separation from the laser-ablated NPs, we observed that the PC NPs dispersant is more toxic than the PET1 NPs dispersant. This can be explained considering that PET1 dispersant does not contain any substances that could cause toxicity [28]. Specifically, the acetic and formic acids traces released in the PET1 water suspension during laser ablation are completely removed by rotavapor treatment. On the other hand, several studies demonstrated that the photo-degradation of PC MPs in the aquatic environment is a source of chemical contamination, owing to the release of monomers and oligomers among which there are also endocrine-disrupting and toxic substances such as Bisphenol A side products, during the aging process [47,59,60,61,62], which could contribute to the toxicity. This hypothesis is supported by preliminary NMR analysis on the PC NPs’ dispersant, which indicates the presence of low-molecular weight intermediate fragments of the polymer chain.

To better understand the effects of NPs on a system closer to the intestinal barrier, differentiated Caco-2 cells have been used as a suitable model for in vitro toxicology studies [41]. On this model, HCS data showed that the NPs do not cause any toxic effects in terms of reduction in cell viability and mitochondrial activity. Differentiated Caco-2 cells were also used for electric cell–substrate impedance sensing measurements (ECIS) [63,64,65]. Data obtained show that all the NPs tested and its dispersants do not cause any damage to the cell monolayer and confirm that even after repeated exposure, barrier integrity is preserved. The absence of toxicity observed can be easily explained as cells become more resistant to external stress when organized in a more tissue-like system with respect to undifferentiated cells [45,66].

To conclude, comparing the toxicological profile of the three NPs investigated using in vitro cell models as Caco-2 and HePG2 cells, PC NPs are the most toxic material, and this could be associated to the by-products obtained by the laser ablation approach. PET2 NPs showed the lower toxic effects, which is most likely associated to the larger size of these colloidal NPs compared to the laser-ablated PET1 NPs. However, on differentiated Caco-2 cells, no toxic effects were observed.

By assessing NPs commonly used, these data can contribute to understanding the potential hazards and risks associated to human exposure to NPs.

## 5. Conclusions

With this study, we aimed to assess the toxicological behavior of commonly used NPs. The results obtained indicate that the toxic effects observed are mainly visible at very high (unrealistic) concentrations of NPs or are linked to the dispersant of the laser-ablated NPs, with higher toxicity for PC, a less extent for PET1, and with even lower effects for PET2 NPs. When NPs were assessed on differentiated Caco-2 cells, all NPs investigated did not show any effects in terms of toxicity. The same was observed by TEER measurements.

The application of different orthogonal methods, including high-throughput approaches, the use of different NPs types and production methods, and the use of different cell models are the basis for an adequate assessment of the biological response to NPs present in the environment and, in this sense, this work contributes to increasing knowledge of possible risks for human health and provides insights for future studies. The development of validated methods and reference materials would contribute to advances in the risk assessment evaluation of NPs.

## Figures and Tables

**Figure 1 nanomaterials-12-01947-f001:**
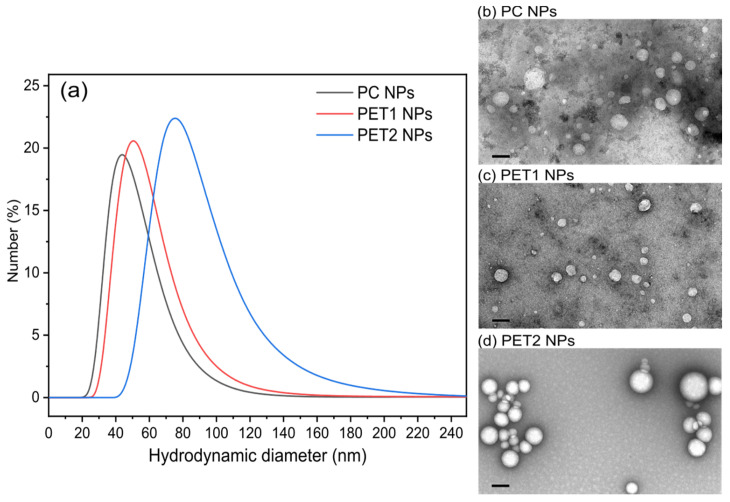
(**a**) Hydrodynamic diameter (D_H_) distribution of PC NPs (black), PET1 NPs (red) and PET2 NPs (blue). Morphology of PC NPs (**b**), PET1 NPs (**c**) and PET2 NPs (**d**) by TEM. Scale bar: 100 nm.

**Figure 2 nanomaterials-12-01947-f002:**
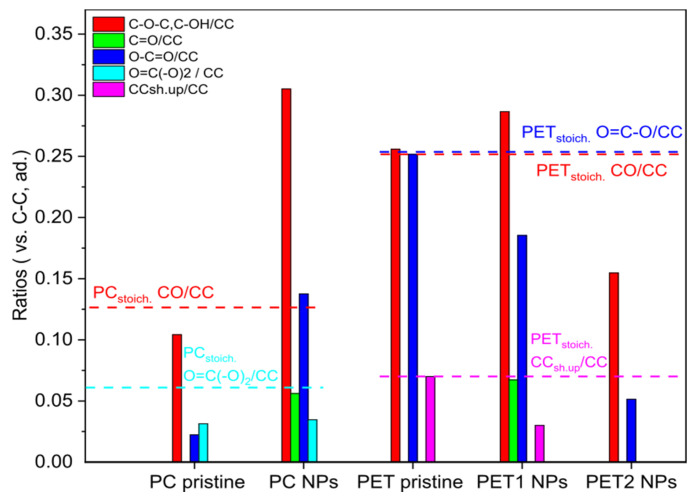
Atomic bonding distribution from XPS C1s spectra analysis. High-resolution spectra curve-fit results from the laser ablated and PET2 NPs. Dotted lines represent stoichiometric reference values.

**Figure 3 nanomaterials-12-01947-f003:**
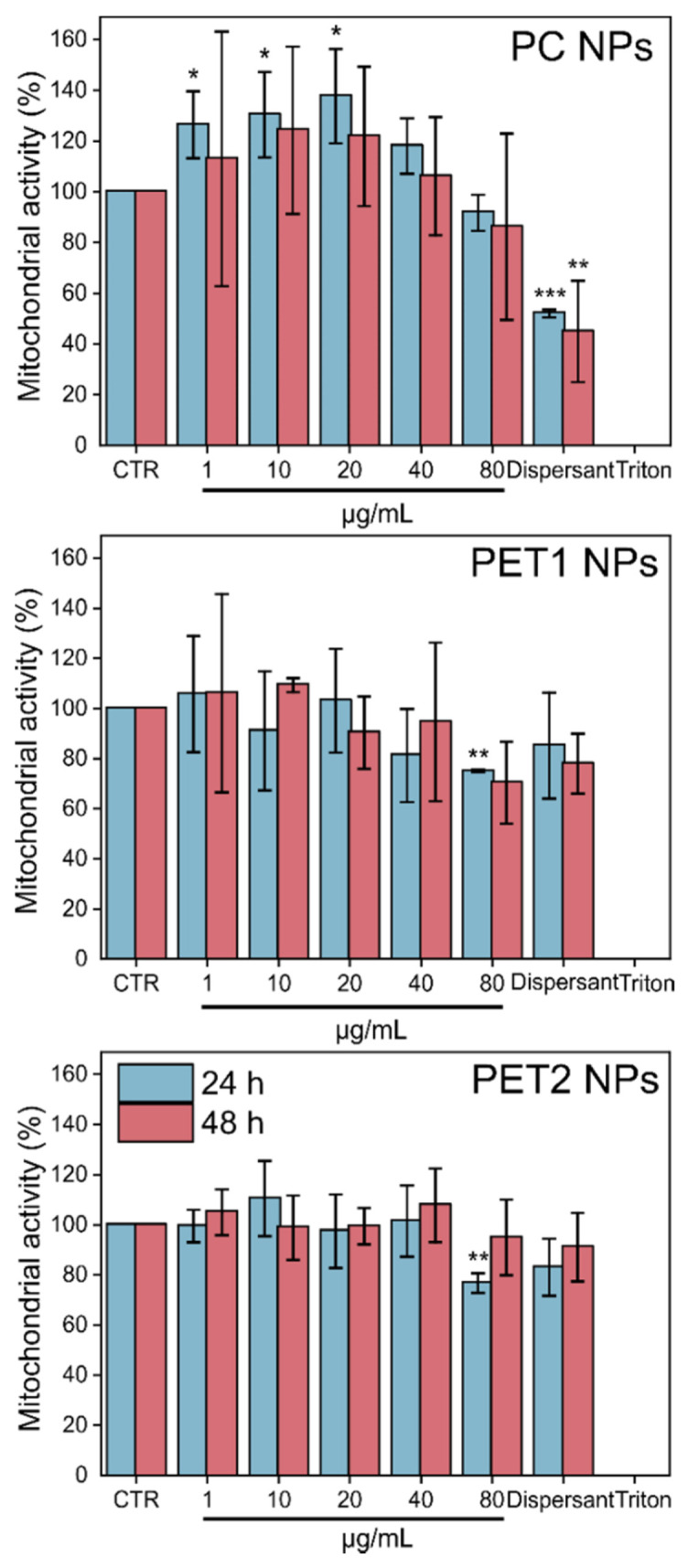
Mitochondrial activity of HePG2 cells after exposure to NPs at the concentrations of 1, 10, 20, 40, 80 µg/mL for 24 or 48 h (MTT assay). Data are reported as average of three independent experiments (each run in technical triplicate) ± SD. *p* < 0.001, *p* < 0.01 and *p* < 0.05 are reported (***, ** and * respectively), calculated versus CTRL (one-way ANOVA).

**Figure 4 nanomaterials-12-01947-f004:**
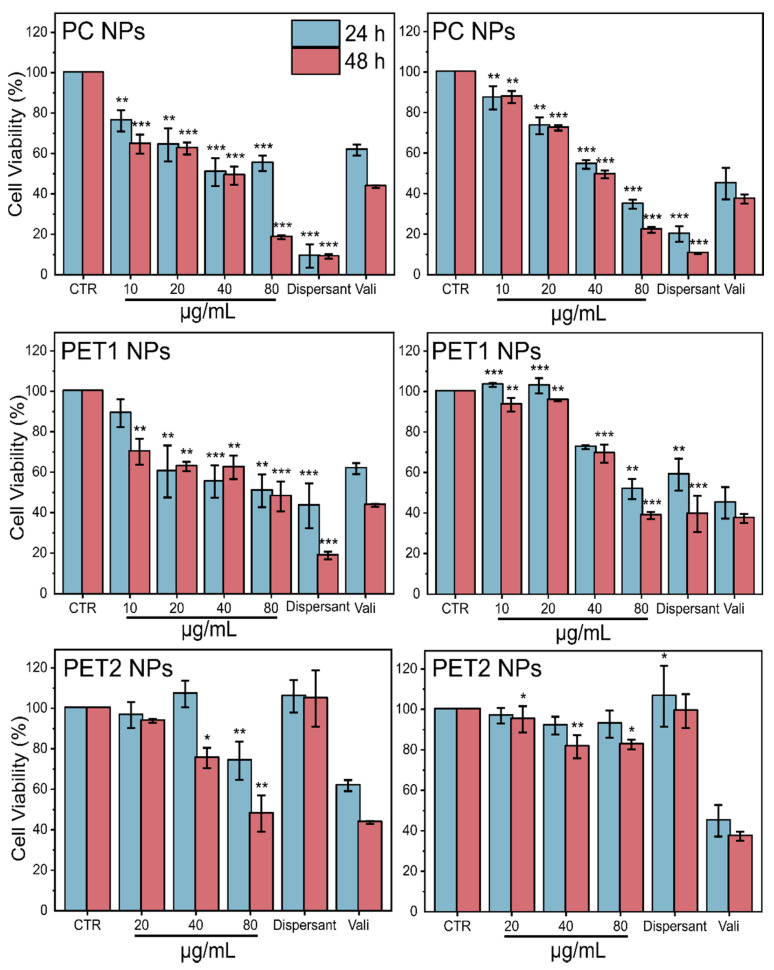
Cell viability of HePG2 and Caco-2 cells exposed to different NPs concentrations or to PC and PET NPs dispersants at 24 or 48 h. Results are expressed as percentage of cell viability compared to the untreated cells. Data are reported as the average of three independent experiments (each run in technical triplicate) ± SD. Valinomycin (900 nM) was used as positive control. *p* < 0.001, *p* < 0.01 and *p* < 0.05 are reported (***, ** and * respectively), calculated versus CTRL (one-way ANOVA).

**Figure 5 nanomaterials-12-01947-f005:**
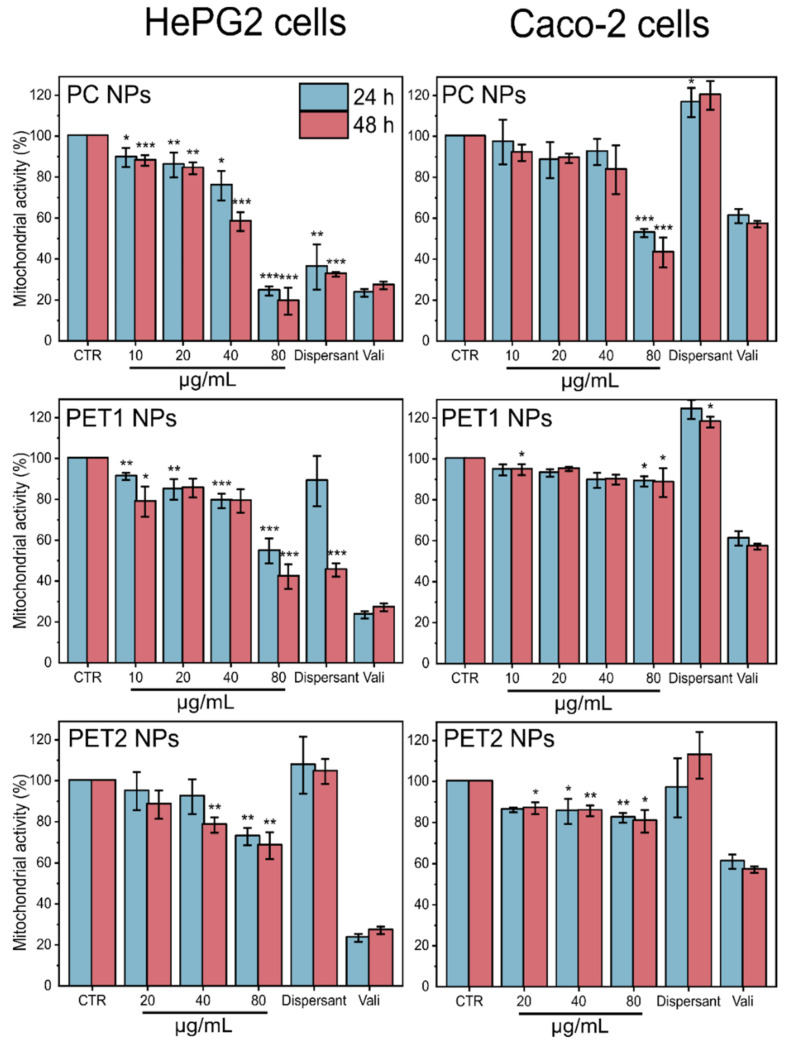
Mitochondrial activity of HePG2 and Caco-2 cells exposed to different concentrations of PC or PET NPs for 24 or 48 h, measured by HCS. Results are expressed as percentage of mitochondrial activity compared to the untreated cells. Valinomycin (900 nM) was used as positive control. Data are reported as the average of three independent experiments (each run in technical triplicate) ± SD. *p* < 0.001, *p* < 0.01 and *p* < 0.05 are reported (***, ** and * respectively), calculated versus CTRL (one-way ANOVA).

**Figure 6 nanomaterials-12-01947-f006:**
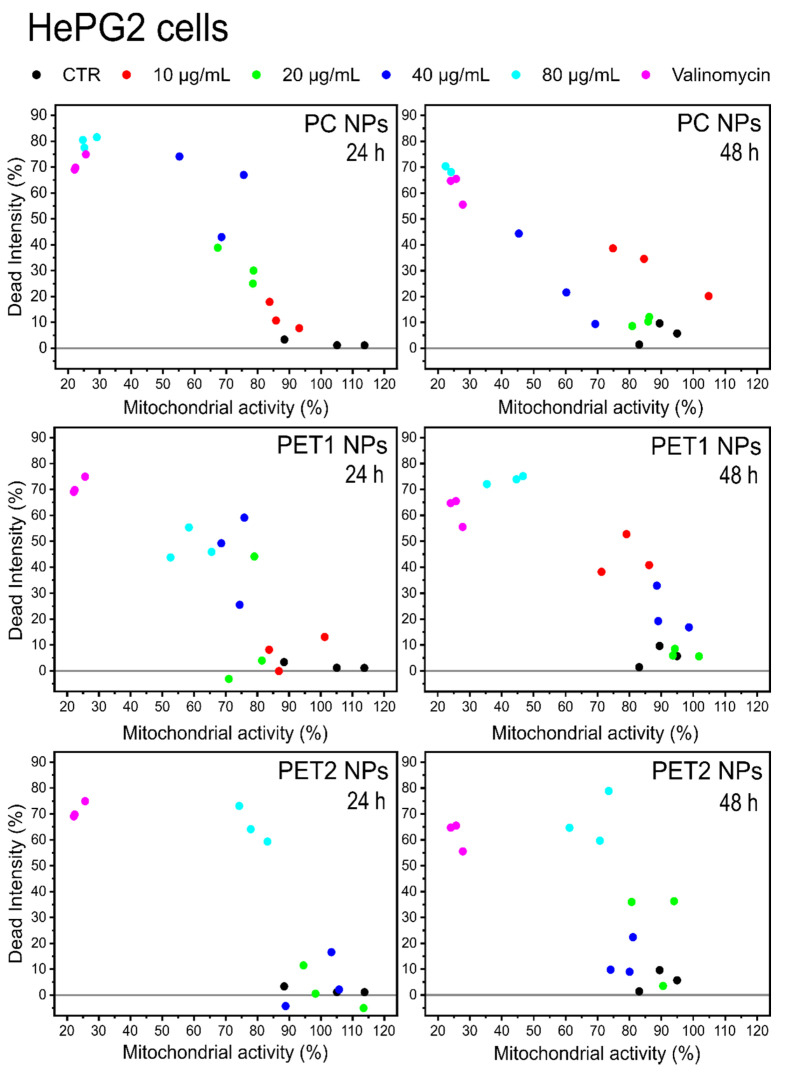
Two-way analysis of HePG2 cells exposed to PC, PET1 or PET2 NPs at concentrations ranging between 10 and 80 μg/mL for 24 or 48 h. Valinomycin was used as positive control. Results are expressed as a function of two different parameters: mitochondrial activity and cellular membrane damage. Data are reported as the average of three independent experiments (each run in triplicate).

**Figure 7 nanomaterials-12-01947-f007:**
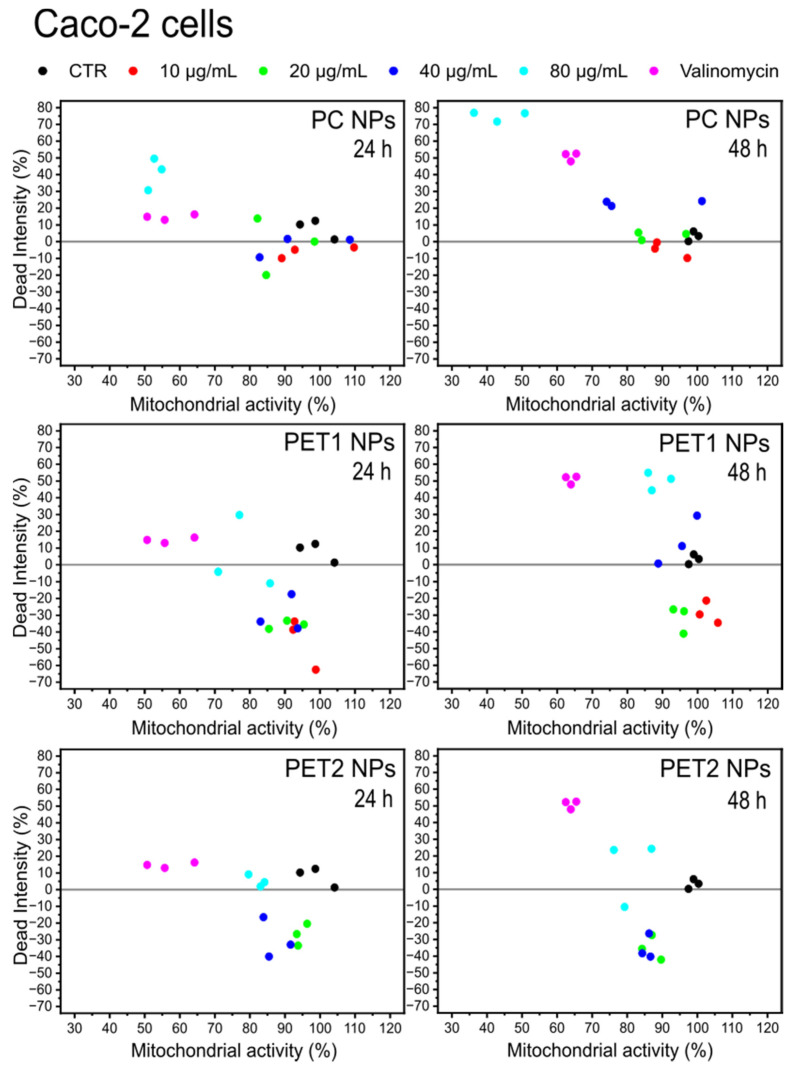
Two-way analysis of Caco-2 cells exposed to PC, PET1 or PET2 NPs at concentrations ranging between 10 and 80 μg/mL for 24 or 48 h. Valinomycin was used as positive control. Results are expressed as a function of two different parameters: mitochondrial activity and cellular membrane damage. Data are reported as the average of three independent experiments (each run in triplicate).

**Figure 8 nanomaterials-12-01947-f008:**
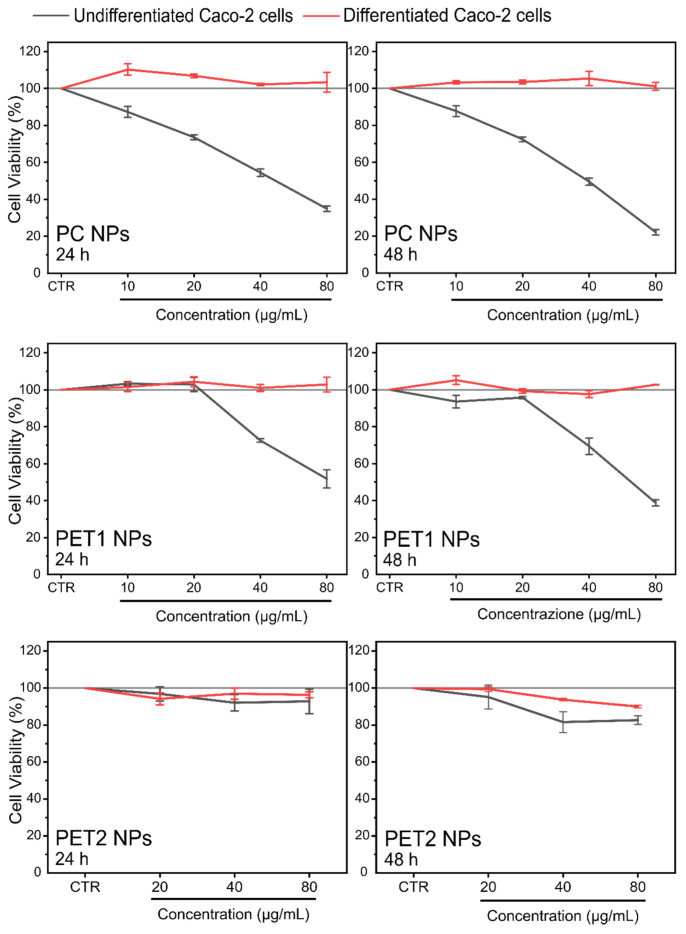
Cell viability of undifferentiated and differentiated Caco-2 cells exposed to different doses of PC, PET1 or PET2 NPs for 24 or 48 h. Data are expressed as percentage of cell viability normalized to the control (untreated cells) and reported as the mean of three independent experiments (each run in triplicate) ± SD.

**Figure 9 nanomaterials-12-01947-f009:**
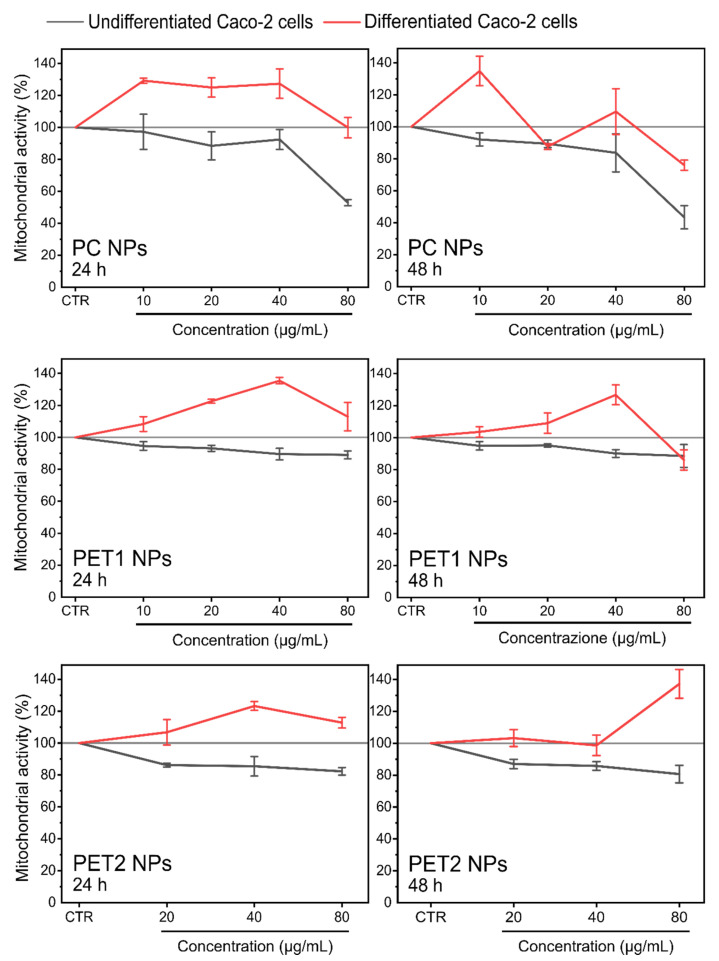
Mitochondrial activity of undifferentiated and differentiated Caco-2 cells exposed to different doses of PC, PET1 or PET2 NPs for 24 or 48 h. Data are expressed as percentage of mitochondrial activity normalized to the control (untreated cells) and reported as the mean of three independent experiments (run in triplicate) ± SD.

**Figure 10 nanomaterials-12-01947-f010:**
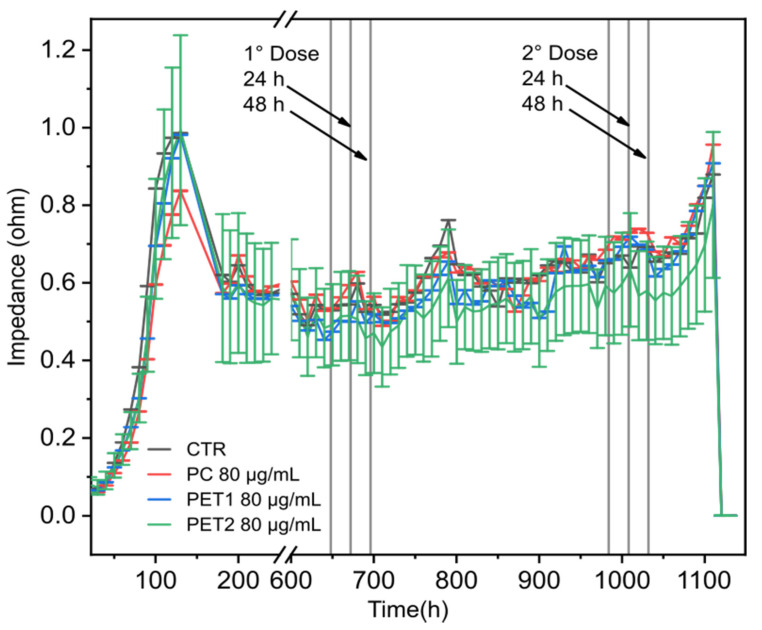
Electrical impedance. Differentiated Caco-2 were exposed to 80 µg/mL of PC, PET1 or PET2 for 48 h; after a recovery period of 12 days, cells were re-exposed to the NPs for another 48 h. At the end of the second treatment, the medium was replaced, and cells were maintained for an additional 48 h. Triton X-100 (0.1%) was then added as positive control.

**Table 1 nanomaterials-12-01947-t001:** The table reports EC_50_ values (µg/mL) for HePG2 and Caco-2 cells exposed to PC, PET1, and PET2 NPs at 24 or 48 h.

	24 h	48 h
**HePG2**	**EC_50_ (** **µg/mL)**	**EC_50_ (** **µg/mL)**
PC NPs	73.12	38.15
PET1 NPs	69.03	68.86
PET2 NPs	74.15	48
**Caco-2**	**EC_50_ (** **µg/mL)**	**EC_50_ (** **µg/mL)**
PC NPs	55.31	44.62
PET1 NPs	79.41	40.06
PET2 NPs	92.45	82.12

## Data Availability

All data related to the study have been presented. They are available at the JRC catalogue: http://data.jrc.ec.europa.eu (accessed on 1 June 2022).

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
