# Peer review of "In Vitro High-Throughput Toxicological Assessment of Nanoplastics"

_nanomaterials, 2022, doi:10.3390/nano12121947_

Round 1
Reviewer 1 Report
Sub-micrometer or nanometer plastic particles have significant health risks, and the toxicological evaluation of plastic particles in mammalian systems is very necessary and important. The manuscript prepared three kinds of nanoplastics by laser ablation and nanoprecipitation, and the toxicological performance on human cell was investigated in detail. It is an interesting work, and the reviewer suggests that the paper can be published only after minor modification as follows.
1) In the last part of Introduction, the introduction of this work should be refined in a paragraph, not three paragraphs.
2) The “3. Result” and “ 4. Discussion” should be consolidated into “ 3 Result and discussion”, and the number of the catalogue should be changed to a three-level title, for example, 1, 1.1 and 1.1.1.
3) The Conclusion is too long, and should be reduced one paragraph.
4) The unit “hours” should be corrected into “h” in Table 1.
5) The possible toxic effect of particle size of nanoplastics on the cell should be analyzed.
6) Expect for PC and PET, the toxic effect of other plastic particles should be discussed, without adding experiments.
Author Response
Dear Reviewer,
The co-authors and I have carefully considered your comments and revised the manuscript accordingly. We thank you.
- In the last part of Introduction, the introduction of this work should be refined in a paragraph, not three paragraphs.
The comment has been taken into account and the introduction has been shorten to one paragraph.
- The “3. Result” and “ 4. Discussion” should be consolidated into “ 3 Result and discussion”, and the number of the catalogue should be changed to a three-level title, for example, 1, 1.1 and 1.1.1.
We have followed the instruction of the journal, which requests separate sections for results and discussion. Nanomaterials | Instructions for Authors (mdpi.com). By personal experience everytime I submitted a paper with result and discussion combined it has been requested to me to separate them. We are willing to make changes if the editor/journal requires it, in addition this comment has not been requested by the other reviewers.
- The Conclusion is too long, and should be reduced one paragraph.
The comment has been taken into account and the conclusion has been.
- The unit “hours” should be corrected into “h” in Table 1.
The change has been made.
- The possible toxic effect of particle size of nanoplastics on the cell should be analyzed.
The effect of particle size on toxicology and cellular uptake is well known and demonstrated for other types of nanoparticles. This has been included in the discussion and further references have been added. Nevertheless, this work is focused on identifying the influence of different methods of synthesis of Nanoplastics “models” and consequently the effects derived from the complexity of more realistic model as the one resulting from the laser ablation process. Although the laser ablation synthesis process produces particles of comparable hydrodynamic diameter in respect to colloidal synthesis (PET2) according to DLS data in terms of intensity (see figure below), the analysis in numbers reflects the high polydispersity of the particles synthesized by laser ablation indicating high amounts in the smaller diameter nanoparticle populations. Polydispersity is a key parameter for a realistic assessment of such pollutants that can hardly be achieved by colloidal synthesis.
6) Expect for PC and PET, the toxic effect of other plastic particles should be discussed, without adding experiments.
We agree with the suggestion, so we have supplemented the discussion. However, it is worth noting that previous studies on NPs have been conducted mainly on commercial NPs or NPs produced by chemical/colloid synthesis that have not shown any toxicity, and as now is explained in the text, toxicological investigations on different models of NPs are lacking to date. In agreement with the referee's comment, this work is among the first to set as its goal the analysis of different models of NPs while also considering different types of in vitro toxicological analysis methods on different cell lines.

Reviewer 2 Report
The manuscript by Tolardo et al." In vitro high-throughput toxicological assessment of nanoplastics" evaluated the toxic effects of nanoplastics using the mammalian in vitro system. Plastic nanoparticles are a severe environmental concern worldwide, and this work addressed an essential issue of estimation of the toxicological effect of three different kinds of nanoplastics. To model environmental nanoplastics, the study used nanoplastics prepared by laser ablation of polycarbonate and polyethylene terephthalate and nanoprecipitation of polyethylene terephthalate. The authors performed comprehensive and detailed physicochemical characterization of the prepared nanoparticles. Assessment of nanoparticlse toxicity was conducted in the mammalian cell lines of intestinal epithelial (Caco-2) and liver (HepG2) origins. The authors employed advanced screening methods, and the presented data would provide a basis for further toxicological studies of nanoparticles.
Novel funding from this work could be of interest to the nanomaterial community. Although the manuscript addresses a significant problem, it has some weaknesses and flaws to address. Here are major and minor comments.
Line 307. Authors have used an advanced high content screening to evaluate the toxicity of nanoparticles in HepG2 and Caco2 cells. It would be beneficial for the study to present representative images of the most effective treatment and positive and negative controls.
Line 340. Please describe more the reason for "two-way analysis". Graphical representation of the correlation between cell death and mitochondrial activity is confusing and does not demonstrate statistical significance.
Line 500. "This hypothesis is supported by preliminary 500 NMR analysis on the PC NPs dispersant, which indicates the presence of low-molecular 501 weight intermediate fragments of the polymer chain." Did authors do this preliminary 500 NMR analysis, but data are not shown, or it is a reference to a published study. Please clarify.
Line 207. "Caco-2 cells were obtained after plating Caco-2 cells in 96-wells ECIS disposable ar-207 rays at a density of 8 x 104 cells/well and maintaining for 21 days till complete differentiation as reported in par 4.1" Apparently, there is a minor typo. Did the authors mean "Differentiated Caco-2 cells were obtained after…."
Author Response
Dear Reviewer,
The co-authors and I have carefully considered your comments and revised the manuscript accordingly. We thank you.
Please see the attachment.

Reviewer 3 Report
Thank you for introducing me to some very interesting research results.
There is no disagreement on the results of the thesis, and we would appreciate it if you could check the English sentences about the thesis once again.
Author Response
Dear Reviewer,
The co-authors and I have revised the manuscript accordingly. We thank you.
Please see the attachment.
